# Is There an Early Morphological Decomposition during L2 Lexical Access? A Meta-Analysis on the Morphological Priming Effect

**DOI:** 10.3390/brainsci13010127

**Published:** 2023-01-11

**Authors:** Ana Isabel Fernandes, Karlos Luna, Ana Paula Soares, Montserrat Comesaña

**Affiliations:** 1Research Unit in Human Cognition, Centro de Investigação em Psicologia, School of Psychology, University of Minho, 4710-057 Braga, Portugal; 2Departamento de Psicología, Universidad Nacional de Colombia, Bogotá 11001, Colombia

**Keywords:** derivational morphology, masked priming effect, non-native and native visual word recognition

## Abstract

A considerable body of experimental data currently exists on the representation and processing of derived words. However, no theoretical account has led to a consensus so far, due in part to inconsistencies in empirical results which show either the presence or the absence of signs of early morphological decomposition during lexical access. In this paper, we present the results of a meta-analysis that sought to examine the robustness of the masked morphological priming effect (MMP) in native and non-native speakers. This effect is indexed by faster responses to targets preceded by morphologically related primes vs. unrelated primes (e.g., fighter-FIGHT < needle-FIGHT), and is perhaps the most widespread effect used to test whether speakers of a given language are sensitive to the morphological components of words at early stages of lexical access. To this end, we selected 10 masked priming lexical decision studies (16 experiments) conducted with native and non-native speakers. Variables such as prime duration and level of L2 proficiency were considered in the analyses to assess their impact on the MMP effect. Results showed significant MMP effects, which were restricted to native speakers. No modulations were found for the prime duration. Results are interpreted in light of prevalent models of complex word processing.

## 1. Introduction

Acquiring a second language (L2) later in life, especially in a classroom setting, is notoriously difficult, particularly when it comes to the acquisition of certain language features, such as derivational morphology, an issue that has been addressed widely in recent decades [1,2,3]. Although a large amount of experimental data on the processing of morphologically derived words is presently available, and a growing number of studies propose theoretical models to accommodate these data, no comprehensive theory has thus far found general approval [4,5]. This may be due, in part, to the presence of controversial and contradictory evidence in the literature, such as the presence or absence of early morphological decomposition during native and non-native lexical access. With the present meta-analysis study, we seek to examine the robustness of the morphological masked priming (MMP) effect, which is, perhaps, the most widespread effect used to test whether native and non-native speakers of a given language are sensitive to morphological word components in an automatic way. Guided by the most influential theories currently being discussed, along with controversial data relating to non-native speakers of various languages, we will identify several variables that might have an impact on the MMP effect, and which might thus have a bearing on the conclusions that can be drawn about the way that complex morphologically derived words are organized and processed in the bilingual mind.

### 1.1. Theoretical Issues of Morphological Processing in First Language Speakers 

The way that suffixed derived words are represented and processed in native languages is not free from controversy [4,6]. Some findings in the literature tend to support the view that words are built from morphemes in the lexicon [5,7]. That is, when real affixes are present in a letter-string, there is an early segmentation of these affixes via a sort of “affix-stripping” mechanism [8]. Thus, an affixed word such as PACKAGE is quickly decomposed into its base and suffix components (PACK + AGE, respectively) before being recognized. Classic models of visual word recognition (e.g., the logogen model) claim that only a word’s base is stored in the input device systems, which match the incoming sensory information with that already stored [9]. Thus, once the word’s base is accessed, the derived word is recognized in the mental lexicon. More recent models assume the existence of a morphological layer between the levels of the form (orthography) and function (semantics/syntactics) of the representation of words, called a lemma [10,11]. According to these models, such as those based on the dual-route connectionist model (DRC; i.e., models in which there are two interconnected routes of processing) [8,12,13], there are two distinct mechanisms involved in morphological processing: i) a whole-word route, by which the meaning of the whole-word is accessed directly; and ii) a morphological segmentation route, by which words are segmented into their morphological components, either through morpho-semantic processing (i.e., the embedded base of a transparent complex derived word such as “fight-er” is stripped—“fight”, and its meaning accessed) or through morpho-orthographic processing (i.e., the affix of a transparent or a pseudo-derived complex word is stripped, and its morphological representation is accessed—e.g., “-er” in the words “fighter” and “corner”). 

Some authors do not acknowledge the existence of morphemes within the lexicon as static units that have to be accessed during comprehension and production. Instead, these authors assert that morphemes can be understood as emergent properties of discriminative learning networks [6,14,15]). For instance, Baayen et al. (2011, 2019) [6,14], in their naïve discriminative learning (NDL) and linear discriminative learning (LDL) models, consider that native speakers, in order to process complex derived words, rely on statistical regularities between the whole word and its base (fighter-fight). That is, our cognitive system would be able to recognize and produce complex derived words when morphemes are not present in their lexicon. More precisely, these authors argue that what appear to be morphological “rules” are the result of recurring semantic and ortho-graphic/phonological patterns, and thus reject morpheme units as distinct linguistic or psycholinguistic constructs [16,17,18]. 

The role of morphological information in lexical access has typically been examined through the assessment of the so-called MMP effect [19,20,21,22,23,24]. This effect is usually obtained with a lexical decision task combined with a masked priming paradigm [25]. This paradigm consists of the presentation of a forward mask (#######), followed by a prime briefly presented in lowercase (around 50 ms; e.g., fighter), which is then replaced by an uppercase target (e.g., FIGHT). Participants are asked to indicate, as quickly and accurately as possible, whether the target is a real word or not. The relationship between prime and target is manipulated, typically using the following conditions: transparent condition, in which primes and targets are semantically, morphologically, and orthographically related (e.g., fighter-FIGHT); pseudo-derived condition (sometimes erroneously called opaque), in which primes and targets are orthographically related and only apparently morphologically related (e.g., corner-CORN); an orthographic control condition, in which primes and targets are only orthographically related (e.g., pillow-PILL); and an unrelated control condition in which primes and targets are not related at all (e.g., driver-FIGHT). Differences in response latencies between unrelated and morphological or pseudo-derived conditions are generally taken as evidence for the existence of an MMP effect, which, in turn, indicates that participants decompose complex derived words into their morphemic constituents before lexical access (or that the sub-lexical distributional properties of a given lexicon are at issue if we consider the tenets of discriminative learning models). This is also substantiated by the presence of differences between morphological conditions (i.e., transparent and pseudo-derived) and orthographic control condition, which indicate that the results are not merely due to orthographic overlap between the prime and the target words.

Results from most masked priming lexical decision studies, comprising complex suffixed derived words and conducted with first languages (L1) have shown that speakers of languages such as English (British [24]; American [22,23]; Australian [19]), French [20], and Italian [21], usually show significant MMP effects for transparent word pairs. The results with pseudo-derived word pairs seem to vary depending on prime presentation time, masking, or task requirements [5]. 

### 1.2. Theoretical Issues of Morphological Processing in Second Language Speakers

Whether MMP effects are also observed in the processing of complex suffixed derived words in a second language (L2), and whether the mechanisms underlying morphological decomposition are similar to those observed in the L1, remains a matter of intense debate [5,26]. Answering these questions regarding the processing of suffixed derived words was the main aim of the present meta-analytic study.

One of the authors who has addressed the representational architecture underlying morphological effects is Ullman [27,28], who developed the declarative/procedural model. According to this model, declarative and procedural memory systems are responsible for different aspects of language acquisition and processing. Whereas declarative memory is linked mainly to lexical processing, procedural memory is strongly associated with grammar processing and language features that are governed by rules such as morphological processing in first language speakers. Contrary, L2 speakers rely more heavily on declarative memory to support both lexical and grammar acquisition, and hence during the processing of suffixed derived L2 words [27]. In this view, then, morphological effects are expected to be weaker in the L2 than in the L1, and the role of L1 transfer on L2 morphological processing is seen to be limited.

Conversely, other authors have argued that L1 and L2 morphological processing share the same mechanisms, and that differences between first and second language speakers can be explained in terms of L1 influence or cognitive resource limitations, rather than lexically based mechanisms [26,29,30]. More specifically, a dual mechanism composed of morpho-semantic and morpho-orthographic routes is proposed to be in operation for L2 speakers, as it is for L1 speakers. Both routes could operate in parallel, in the case of transparent words, but not in the case of pseudo-derived words, where only morpho-orthographic processing is required. 

In order to test the above-mentioned theoretical proposals, Silva and Clahsen (2008; experiments 3 and 4) [31] used a masked priming lexical decision task in English (with a 60 ms prime duration) to compare the performance of speakers of English as L1 with that of late-acquiring, highly proficient speakers of English (L2) with either Chinese or German as L1. Three conditions were used: transparent (e.g., neatness-NEAT), identity (presentation of the same word as prime and target, e.g., neat-NEAT), and unrelated (dark-NEAT). The authors observed a significant MMP effect for L1 speakers (i.e., shorter reaction times for transparent than for unrelated conditions, with no differences between identity and transparent conditions) and a partial priming effect for L2 speakers (i.e., significant priming effects for transparent and identity conditions, although higher for the later). These findings led the authors to conclude that L1 and L2 speakers use the morphological structure while processing suffixed derived words. Although interesting, this study has significant limitations that may have influenced the results. For instance, only primes containing a specific suffix (i.e., “-ness” or “-ity”) were used; also, there was no orthographic control condition (e.g., brothel-BROTH), and such control can be useful in indicating whether the observed effect of morphological priming is due to morphological processing rather than an orthographic overlap between primes and targets words. Nevertheless, several subsequent masked priming studies involving greater control of possible explanatory variables (e.g., the inclusion of an orthographic control condition in the experiment), and again looking at L2 speakers, observed a similar performance between L1 and L2 speakers of different populations (with late-acquiring, highly proficient Russian–German speakers [32]; late-acquiring, highly proficient Polish–German speakers [33]; with early acquiring, highly proficient Spanish–English speakers, and late-acquiring, highly proficient Dutch–English speakers [30]; with late-acquiring, highly proficient Russian–German speakers [34]; with late-acquiring, highly proficient English–German speakers [35]). 

For example, Diependaele et al. (2011) [30] compared L1 speakers of English to highly proficient, early acquiring Spanish–English and late-acquiring Dutch–English speakers, using a masked priming lexical decision task in English with a 53 ms prime duration. The following four conditions were used: transparent (e.g., viewer-VIEW) and pseudo-derived (corner-CORN) morphological conditions, orthographic (e.g., freeze-FREE), and unrelated (e.g., penalty-VIEW) control conditions. Results showed significant MMP effects for both transparent and pseudo-derived conditions in L1 and in early and late L2 speakers, with orthographic and unrelated control conditions statistically differing from transparent and pseudo-derived ones. These findings support the view that L2 speakers process the morphological information of complex suffixed derived words as L1 speakers do [26,29,30].

However, some empirical investigations with late L2 speakers with high-to-intermediate proficiency (with German-English speakers [36]; with L2 speakers of Turkish [37]; with Italian–English speakers [38]) observed that only L1 speakers seem to decompose a complex suffixed derived word into its morphemic constituents, supporting Ullman’s (2005) proposal [27]. More precisely, Heyer and Clahsen (2015) [36] applied a masked prime lexical decision task, with a longer prime duration (67 ms) relative to Diependaele et al. (2011—53 ms) [30], with a transparent morphological condition (e.g., darkness-DARK) as well as an orthographic (e.g., example-EXAM) and an unrelated (e.g., traffic-DARK) control conditions, to L1 speakers of English and highly proficient German–English speakers. Significant MMP effects were observed in both groups of participants. However, a significant priming effect for the orthographic condition also emerged in the L2 group, with no statistically significant differences seen between transparent and orthographic control conditions. These results led the authors to conclude that L2 speakers, while processing complex suffixed derived words, rely more on the orthographic relationship between primes and targets, and do not decompose stimuli of this kind into their morphemic constituents, as L1 speakers do. 

In sum, the existing literature on suffixed derived words, which uses the masked priming lexical decision task with L2 speakers, shows mixed results. Indeed, some works support Ullman’s (2005) [27] proposal of less automatization of morphological processing in L2 [36,37,38] and others show the presence of similar morphological decomposition mechanisms in L1 and L2 [30,32,33,34,35]. 

#### Variables Affecting MMP Effects 

The divergences observed in the literature may be explained in terms of methodological (i.e., the type of prime–target word pairs used; duration of prime’s presentation) and linguistic variables (i.e., L2 proficiency), which will be discussed through this section and considered in the meta-analysis.

Regarding the selection of the type of prime–target word pairs, the most commonly used sets of experimental conditions with L1 speakers, besides the unrelated condition, are the following: transparent (e.g., fighter-FIGHT), pseudo-derived (e.g., corner-CORN) and orthographic control (e.g., pillow-PILL) [19,20,21,22,23,24]. As has already been mentioned, the presence of significant MMP effects for both transparent and pseudo-derived morphological conditions (which differ from the orthographic control condition) can be seen as the result of the automatic and faster processing of the morphological information of complex derived words before lexical access. However, not all studies examining L2 morphological processing have used these three conditions; some use a selection of them [36], or indeed alternative ones (e.g., a semantic condition, in which only a semantic relationship is present between prime and target as in doctor-PHYSICIAN and/or and identity condition, fight-FIGHT). This happens, for instance, in the studies of Silva and Clahsen (2008) [31] and Jacob et al. (2017) [34]. This variation hampers direct comparisons between studies, and may also explain, at least in part, the inconsistent conclusions which have been drawn. 

Furthermore, it seems that the duration of a prime’s presentation modulates the morphological processing of suffixed derived words in L2 speakers. Zhang et al. (2016) [39] carried out a masked priming lexical decision task in English with a group of late-acquiring, intermediate-proficiency Chinese–English speakers while manipulating the duration of prime presentation (40 and 80 ms). Four experimental conditions were used: transparent (e.g., reporter-REPORT), pseudo-derived (e.g., corner-CORN), semantic (e.g., choice-SELECTION), and an unrelated (e.g., north-DRESS), to guarantee that the priming effect in the transparent condition was not due to semantic overlap. The authors observed a significant morphological priming effect for the transparent condition when the prime duration was short, and a significant priming effect for the pseudo-derived condition when the prime duration was longer. No priming effect was observed in either prime duration for the semantic control condition. The authors explain this variance in the pattern of the MMP effect by pointing out that whereas for transparent words, morpho-semantic and morpho-orthographic processing occurs, for pseudo-derived words only morpho-orthographic processing is present, which may lead to a later priming effect. According to Zhang et al. (2016) [39], these results are in line with dual-route connectionist models [40], according to which both morpho-semantic and morpho-orthographic routes are required while processing complex derived words. However, it is important to note that we would expect significant priming effects to occur earlier during visual word recognition for the pseudo-derived condition, as has been observed in other studies with non-native speakers with a shorter prime duration (e.g., Diependaele et al., 2011 [30], with a prime duration of 43 ms). It is thus likely necessary to consider that these results might have been affected not only by prime duration but also by the degree of L2 proficiency (intermediate) of the participants in the study of Zhang’s et al. (2016) [39], and hence that different results might be found in L2 speakers with higher levels of proficiency. 

Indeed, proficiency is another variable that seems to modulate the MMP effect. More specifically, Li et al. (2017) [41] applied a masked priming lexical decision task in English with transparent (e.g., hunter-HUNT), pseudo-derived (e.g., corner-CORN), orthographic control (e.g., freeze-FREE) and unrelated (e.g., illness-HUNT) conditions (50 ms of prime duration). Speakers of English as L1 and L2 speakers with high vs. low English proficiency (their L1 was Chinese) took part in the study. These authors observed a similar pattern of results between L1 speakers of English and the highly proficient L2 speakers (i.e., significant priming effects for transparent and pseudo-derived conditions, with no significant priming effect for the orthographic control condition); by contrast, low-proficiency L2 speakers showed a different pattern of results (i.e., significant priming effects for transparent and orthographic conditions, but no significant priming effect for the pseudo-derived condition). The study suggests that only L2 speakers with high levels of L2 proficiency are able to process the morphological information of L2 suffixed derived words in the way that L1 speakers do. This may also explain the reliance on the orthographic information that intermediate-proficiency L2 speakers who took part in the study by Viviani and Crepaldi (2019) [38] showed. 

Furthermore, even though most of the experimental studies mentioned in the previous section stated that their participants consisted of L2 speakers with high proficiency [30,32,33,34,35,36,37], the same measures and criteria to evaluate proficiency were not always used. For instance, most of them used a standard test to assess their participants’ proficiency [31,32,33,34,35,36,37]. They classified their sample’s proficiency based on the average score of all participants, which is not an exact measure since there is some variance when one looks at the standard deviation reported in some studies [32]. Other, however, have considered subjective measures taken from socio-linguistic questionnaires developed by the authors [30] or a combination of both subjective measures (e.g., participants’ daily use of L2 or the exposition to it) and standardized tests [39,41]. In addition, some have considered different measures (e.g., spelling, vocabulary, phonemic fluency, morphological awareness, oral and reading comprehension) to determine their participants’ L2 proficiency [38]. Considering the above, exploring the effect of L2 proficiency on the MMP effect seems very relevant at present.

In sum, different variables, such as experimental ones (e.g., duration of prime’s presentation [39]) or individual ones (e.g., L2’s proficiency [41]), need to be considered when conducting masked priming experiments with L2 speakers in order to arrive at conclusive results regarding the way complex derived words are represented and processed.

### 1.3. The Present Study

As noted above, the results of the reviewed empirical studies conducted with L2 speakers are inconsistent, with some evidence supporting the claim that L1 and L2 speakers are alike in processing morphological information of complex suffixed derived words [30,31,32,33,34,35], and other evidence showing that L2 speakers rely more on lexical information while processing words of this type [36,37,38]. Such inconsistencies might be explained in terms of the influence of different variables which thus far have not been considered in such studies. 

With this in mind, a meta-analytic study was conducted in order to shed light on still unresolved issues relating to the nature of the representation and processing of suffixed derived words. More precisely, we intend to address how complex suffixed derived words are processed by L1 and L2 speakers of different linguistic populations by examining the robustness of the MMP effect itself, as well as through a quantitative analysis of both the role of prime duration and also the level of L2 proficiency, which seems to modulate the MMP effects, as suggested by Zhang et al., (2016) and Li et al., (2017)’s studies. 

Considering the results of the studies under review, we expect the following findings: (i) a small-to-moderately sized MMP effect in L2 speakers that supports the use of morphological structure during lexical access (or as evidence of form and meaning representations whose development is driven by discriminative learning networks); (ii) no differences in the size of priming effect between the transparent and pseudo-derived conditions (but differences between these two conditions and the orthographic condition), at least for L1 and L2 speakers with high L2 proficiency; and (iii) an increase in the size of the MMP effect as L2 proficiency increases.

## 2. Materials and Methods

A literature search was conducted on the Web of Science and Google Scholar databases. Considering the above-cited goals, we selected the following three keywords: “bilinguals” (this term allowed us to specify the population we wanted to study, comprising both balanced and unbalanced bilinguals since we wanted to explore the effect of L2 proficiency in the MMP); “derivation”—(to specify the morphological type of words we wanted to focus on); and “morphological priming” (to limit our search to empirical studies which have used a masked priming lexical decision task). A total of 323 papers were obtained. From these, 16 were selected (one of which was an unpublished PhD. thesis), based on the following criteria: (1) a masked priming lexical decision task was used with a group of L2 speakers of a given language, and (2) the experimental stimuli consisted of suffixed complex derived words (see Figure 1).

From the 16 selected papers, 6 were excluded because they did not report enough statistical information to perform the meta-analysis (we contacted the authors, but they did not reply). Sixteen experiments were identified in the remaining 10 articles (all published, except for that by Otto, 2012 [42]), with a total of 87 response time comparisons between the related and the unrelated control conditions. Hedges’ g statistics were computed for each comparison (see Appendix A, which consists on a forest plot with effect sizes and descriptive data for each study here: https://osf.io/4stgu/?view_only=7c47dd3ca4304d0ba6965e8e827a96eb accessed on 4 April 2022). None of the reports included the correlation between measures, which is needed to compute the variance of g. We repeated the analyses with the variance computed from correlations equal to 0.1, 0.5, and 0.9. The results showed only trivial differences, which did not affect interpretation. Below we report the results with a correlation of 0.5 between measures and the descriptive with correlations of 0.1 and 0.9 in the Appendix A. We computed effect sizes as the response time of the related priming condition minus the response time of the unrelated priming condition (control). Thus, a positive g shows priming effects (e.g., faster responses for the related priming condition), whereas a negative g indicates faster responses in the unrelated condition. Moreover, L2 speakers’ data were compared to the data of L1 speakers present in 9 of the 10 selected papers.

Effect sizes in a meta-analysis must be independent, but the same sample provided several comparisons of interest in all cases. For example, an experiment with a morphological priming condition that included both transparent and pseudo-derived conditions provided two valid comparisons against a control. We used the robust variance estimation method to avoid problems related to the non-independence of the effect sizes [43]. This method considers the correlation between observations and was implemented using the *robumeta package* [44] for R [44,45] (R Core Team, 2018). *Robumeta* includes a parameter (r) for the within-study correlation between effect sizes, and a tool to check the sensitivity of the outcome to different values of r. This is achieved by calculating the meta-analytic estimate when r varies on several levels from 0 to 1. In our sample of effect sizes, variations in r caused only trivial changes in the results and did not affect interpretation. We report the results with the default parameter r = 0.80. All the confidence intervals are 95% CI.

## 3. Results

Results are organized as follows: first, we analyzed the data of the whole sample with L2 and L1 speakers. Then, subsequent analyses were conducted to examine: (i) if there were differences between them; (ii) the contribution of the variables reviewed in the introduction that may be affecting the MMP effect (prime duration and L2 proficiency [the analysis of this last variable was restricted to the L2 speakers’ group]).

The meta-analysis with all 87 effects showed a small but significant morphological priming effect, g = 0.175, SE = 0.022, CI [0.130, 0.221], t(18.65) = 8.07, *p* < 0.001. To test our specific hypothesis regarding the effect of the prime-type relationship (i.e., priming effects for transparent and pseudo-derived words compared with control conditions), we computed a series of meta-analyses. The prime-type relationship had more than two levels. In that case, the *robumeta* package does not report an omnibus test; thus, we report pairwise comparisons between levels. For all the analyses reported we used the correction for small samples, as recommended by [46].

To be more precise, the prime-type relationship included five levels: transparent (k = 26), pseudo-derived (k = 11), identity (k = 22), orthographic (k = 20), and semantic (k = 8). The three latter ones worked as controls. Estimates for the five conditions are presented in Table 1, first row. All of them were higher than zero (all *p* < 0.004) except for the semantic condition (*p* = 0.366). We applied the Bonferroni correction for comparisons between conditions and set alpha to 0.005 (10 comparisons). The priming effect was highest in the identity condition (although it did not differ from the transparent condition); then came the transparent and pseudo-derived conditions, with no differences between them; the lowest estimates were for the orthographic and semantic conditions, and also had no differences between them. In addition, there were no differences between the pseudo-derived and the orthographic conditions. We report the statistical comparisons in Appendix A (https://osf.io/4stgu/?view_only=7c47dd3ca4304d0ba6965e8e827a96eb, accessed on 4 April 2022).

Having observed MMP effects, two subsequent analyses were conducted to explore the effect of prime type in L1 and L2 speakers. Estimates for L1 and L2 speakers per prime-type relation condition are presented in Table 1, and statistical comparisons are reported in Appendix A (https://osf.io/4stgu/?view_only=7c47dd3ca4304d0ba6965e8e827a96eb, accessed on 4 April 2022). For studies with L1 speakers (k = 39), estimates were higher for the identity (k = 11) and transparent conditions (k = 12), with no differences between them, than for the pseudo-derived (k = 4), orthographic (k = 9), and semantic conditions (k = 3), also with no differences between them. There were also no differences between the transparent and pseudo-derived conditions, and only the transparent condition was different from the orthographic and semantic control conditions. For L2 speakers (k = 48), the estimate was higher for the identity condition (k = 11) than for the transparent (k = 14), pseudo-derived (k = 7), orthographic (k = 11), and semantic conditions (k = 5), with no differences between the last four. Thus, despite the graded priming effect across conditions at a descriptive level, there were no differences between estimates. Hence, we cannot say that L2 speakers process morphological information of complex derived words before lexical access. Note, however, that the sample of L2 speakers is more heterogeneous than that of L1 speakers due to differences in levels of L2 proficiency. In order to evaluate the impact of this variable on the results, as well as that of prime duration, further analyses were conducted. 

That is, we turned to the question of whether the effect was modulated by group (L1 speakers vs. L2 speakers, as the analysis above, seems to indicate), prime duration (equal or shorter than 50 ms, or longer than 50 ms), or, for L2 speakers, participants’ L2 proficiency (high, intermediate, or low). In order to see more clearly the impact of these variables on the size of the effect, we ran separate analyses for the transparent and pseudo-derived conditions. We also analyzed the orthographic condition to obtain a better picture of the role of orthographic overlap in L2 speakers (bear in mind that several studies showed differing degrees of involvement of this variable as a function of L2 proficiency, [41]). For the semantic condition, the analyses were uninformative because there were only eight effects. Thus, the semantic condition was not analyzed further. We first report the results of the moderators for the transparent condition, then for the pseudo-derived condition, and finally for the orthographic condition. 

When effects from the transparent condition were analyzed, L1 speakers showed stronger priming effects, g = 0.257, SE = 0.029, CI [0.192, 0.321], than L2 speakers, g = 0.136, SE = 0.035, CI [0.059, 0.214], t(17.66) = 2.53, *p* = 0.021. Both effects were higher than zero (lowest *p* = 0.003). Prime duration did not affect the morphological priming effect: the effects of a short duration, g = 0.174, SE = 0.081, CI [−0.054, 0.403] and long duration, g = 0.192, SE = 0.023, CI [0.144, 0.240] were similar, t(5.64) = 0.21, *p* = 0.841. Only the effect for a long duration was different from zero (*p* < 0.001). As for proficiency, there was only one effect for low proficiency and thus that level was excluded from the analysis. The effect was numerically higher for high proficiency, g = 0.168, SE = 0.032, CI [0.096, 0.240], than for medium proficiency, g = −0.015, SE = 0.102, CI [−0.505, 0.474], but the difference was not significant, t(3.74) = 1.72, *p* = 0.165. Only the effect for highly proficient participants was different from zero (*p* < 0.001). To further test the effect of proficiency, we also compared the effect of L1 speakers, g = 0.260, SE = 0.027, CI [0.199, 0.321] against L2 speakers with high proficiency, g = 0.162, SE = 0.034, CI [0.082, 0.241], and found a higher effect in the former than in the latter, t(14.89) = 2.28, *p* = 0.038. Both effects were higher than zero (higher *p* = 0.002). When comparing the effect for L1 speakers, g = 0.255, SE = 0.026, CI [0.197, 0.313] and L2 speakers with medium proficiency, g = 0.026, SE = 0.086, CI [−0.342, 0.395], the numerical difference was large but it was not significant, probably due to the low number of studies in the medium-proficiency condition (k = 3), t(3.17) = 2.47, *p* = 0.086. Only the effect for L1 speakers was higher than zero (*p* < 0.001; for non-native speakers with medium proficiency, *p* = 0.787).

There were 11 effects in the pseudo-derived condition, and thus the analyses reported in this section should be interpreted with caution. As per language, the MMP effect was similar for L1 speakers, g = 0.119, SE = 0.037, CI [−0.017, 0.255] and L2 speakers, g = 0.101, SE = 0.014, CI [0.064, 0.137], t(3.99) = 0.48, *p* = 0.659. Because of the lower standard error for L2 speakers, that effect was different from zero (*p* < 0.001), but it was not different for L1 speakers (*p* = 0.067). Since there was only one effect with a long prime duration, the analysis of prime duration was not informative. As per proficiency, there was only one effect with low proficiency, and thus that level was excluded. The morphological priming effect was similar for high-, g = 0.105, SE = 0.024, CI [−0.003, 0.212], and for medium-proficiency participants, g = 0.088, SE = 0.019, CI [−0.004, 0.179], t(3.59) = 0.56, *p* = 0.606. Both effects showed a non-significant trend towards differences from zero (*p* = 0.052 and *p* = 0.054, respectively). For participants with high proficiency, there were no differences between L1, g = 0.115, SE = 0.028, CI [−0.007, 0.236], and L2 speakers, g = 0.107, SE = 0.017, CI [0.017, 0.197], t(2.82) = 0.28, *p* = 0.799. Only the second effect was different from zero (*p* = 0.038); for L1, there was only a non-significant trend (*p* = 0.055). Similarly, for participants with medium proficiency, there were also no differences between L1, g = 0.104, SE = 0.037, CI [−0.018, 0.225], and L2 speakers, g = 0.086, SE = 0.027, CI [−0.029, 0.201], t(3.74) = 0.38, *p* = 0.726. Again, both effects showed non-significant trends toward differences with zero, *p* = 0.073 and *p* = 0.084, respectively.

When the effects from the orthographic priming condition were analyzed, the priming effect was higher for L2 speakers, g = 0.106, SE = 0.026, CI [0.046, 0.166], than for L1 speakers, g = 0.015, SE = 0.026, CI [−0.047, 0.077], t(12.96) = 2.44, *p* = 0.030. Only the former effect was different from zero (*p* = 0.003). As per prime duration, there were no differences between short durations, g = 0.064, SE = 0.023, CI [0.014, 0.115], and long durations, g = 0.085, SE = 0.044, CI [−0.475, 0.645], t(1.17) = 0.42, *p* = 0.741. Again, due to the different standard errors between conditions, only the effect for the short duration was different from zero (*p* = 0.017; for long *p* = 0.305). There was only one effect for medium and low proficiencies, and thus the analysis was not conducted.

## 4. Discussion

The present meta-analysis aimed to examine the robustness of what is perhaps the most commonly used indicator of automatic morphological priming during L1 and L2 visual word recognition: the masked morphological priming (MMP) effect. Variables such as L2 proficiency and prime duration were considered to assess their impact on the effect. Results were clear-cut, as only L1 speakers were sensitive to the morphological structure of suffixed derived words, which was consistent with what was observed in some previous empirical studies [36,37,38] - see also the study of Lõo et al., 2020 [47], for more recent evidence. In fact, L1 speakers showed no differences, either between transparent and identical prime–target pairs (note these conditions differed from the control conditions: orthographic and semantic) or between the transparent and pseudo-morphological ones (though the pseudo-derived condition was not different from the control conditions), which goes in line with most existing empirical studies with L1 speakers [19,20,21,22,23,24]. These findings observed in L1 speakers fit well with DRC models, which hold that complex derived words are processed through two main routes: a whole-word representation route; and a decomposition route, where both morpho-semantic and morpho-orthographic processing occurs [8,12,13]. This constitutes evidence for rapid morpheme extraction independently of orthographic and semantic relatedness (see Royle and Steinhauer, 2021 for an excellent review of converging neuropsychological evidence). However, even though only the transparent condition was differentiated from control conditions, no differences were found between transparent and pseudo-derived conditions, and hence the data may be also consistent with those accounts which claim that masked morphological priming effects are due to statistical regularities between the prime and the target, with no morphological mechanisms of decomposition occurring [6,14,48], or to consistencies between orthography and semantics (the so-called orthography–semantics consistency [OSC] effect) [17]. More research is needed to disentangle these two proposals. It is worth noting here that the differences that have been observed in some masked priming lexical decision studies between transparent and pseudo-derived conditions may be explained by individual linguistic differences such as a semantic vs. orthographic profile [19], or a faster vs. slower reader profile [49], rather than by a greater reliance on lexical over morphological information. More precisely, Andrews and Lo (2013) [19] tested Australian English speakers in a masked priming lexical decision task with a 50 ms prime duration. Four conditions were included: transparent, pseudo-derived, orthographic, and unrelated conditions. Participants were divided into two linguistic profile groups according to their performance in spelling and vocabulary measures: participants with higher scores in spelling tasks were allocated to the orthographic profile group, while participants with higher scores in vocabulary tasks were allocated to the semantic profile group. Through this procedure, the authors observed that both groups of participants seemed to be sensitive to the morphological information of complex suffixed derived words (i.e., significant priming effects for transparent and pseudo-derived conditions, with no priming effect for the orthographic control condition). However, when the linguistic profile variable was included in the analyses, a different pattern of results emerged. Although both linguistic profiles showed a similar MMP priming effect for the transparent condition, differences emerged for the pseudo-derived condition, with the effect being stronger for participants with an orthographic linguistic profile. Additionally, Medeiros and Duñabeitia (2016) [49] examined the performance of Spanish speakers on a masked priming lexical decision task (50 ms of prime duration) with related conditions (i.e., the prime and target shared the same suffix, e.g., *herrero-BASURERO*) and unrelated ones (i.e., the prime and target did not share the same suffix, e.g., *adultez-BASURERO*). Participants were allocated to two different reading profile groups: “faster readers” (short reaction times during reading) and “slower readers” (long reaction times). The authors observed a significant suffix priming effect, but this was restricted to the “slower readers”. Thus, these variables seem to modulate the way complex words are processed and should be taken into consideration in future masked priming studies with L1 and L2 speakers. 

Moreover, the size of priming effects in the transparent and pseudo-derived conditions increased numerically with language proficiency (L1 > high L2 proficiency > intermediate L2 proficiency), following what was observed by Li et al. (2017) [41]. However, L2 speakers were more sensitive to orthographic than morphological relatedness. These findings support theoretical proposals such as the one developed by Ullman in 2005 [27], in which it is argued that L1 and L2 morphological processing do not share the same underlying mechanisms, at least when L2 speakers learn the L2 late in life and/or are unbalanced bilinguals. Indeed, these findings sustain the idea that L2 speakers rely more heavily on declarative memory to support lexical and grammar acquisition and to process L2 suffixed derived words. This may be the reason why MMP effects did not differ from the orthographic control condition. Another possible explanation for the different pattern of results observed between L1 and L2 speakers could be the one provided by Chuang et al. (2020) [50]. The authors argue that less exposure of the latter group of participants to L2 complex derived words, especially the lower frequent ones, would make it harder for them to generalize derivational rules of their L1 language to new L2 complex derived words and recognize them as such.

If Ullman (2005) and Chuang et al. (2020) [27,50] are right, then differences between L1 and L2 speakers should not be observed when bilinguals learn the two languages early in life and thus are balanced bilinguals, an issue to be examined in future research. Unfortunately, studies with balanced bilinguals are not as common as those with L2 unbalanced speakers, probably because the latter group is far more commonly found in the population. Indeed, all the studies included in the present meta-analyses tested L2 speakers who learned the L2 later in life. As far as we know, the only study that examined MMP effects on early acquiring and probably balanced bilinguals is that of Veríssimo et al. (2017) [51], who did not find modulations on the MMP effect as a function of the age of L2 acquisition. This study, however, was not included in the present meta-analysis because we did not have access to the raw data. In any case, more effects than those from a single study would be needed to run analyses.

Furthermore, differences between L1 and L2 speakers were not modulated by prime duration. In fact, this variable failed to impact the MMP effect. This is consistent with what Marslen-Wilson et al. (2008) [22] have found, that is, significant MMP effects regardless of prime duration (33 ms, 48 ms, and 72 ms) in a sample of speakers of English catageorised as L1. This contrasts, though, with the results of Zhang et al.’s (2016) [39] study with late-acquiring Chinese–English speakers with an intermediate level of English proficiency, where a different pattern of results was found in shorter (40 ms) and longer prime durations (80 ms). We should bear in mind, however, that the results of Zhang et al. (2016) [39] may reflect the effect of participants’ lower L2 proficiency, as suggested by the results of Li et al. (2017) [41] and by the present results since the MMP effect observed with transparent pairs was higher for highly proficient speakers when comparing with intermediate ones. 

Additionally, it is also important to notice that the results observed with the L2 speaker group may also reflect the influence of participants’ L1 knowledge on their respective L2s, i.e., the transfer (or not) of derivational morphological knowledge of L1 to L2 [3,52]. In most of the empirical studies that entered our meta-analysis, both L1 and L2 were rich in derivational morphology [32,33,34,35,36,38,42], so it is possible for this transfer from L1 to L2 to occur. However, the same thinking cannot be applied to the studies performed with Chinese(L1)–English(L2) L2 speakers [39,41], since the Chinese language is not as rich as English when it comes to derivational morphology [52]. Thus, it is important to consider the morphological aspects and characteristics of L1 in future investigations on derivational morphological priming with L2 speakers.

## 5. Conclusions

To summarize, the present meta-analytic study allows us to affirm that, although the size of priming effects in the transparent and pseudo-derived conditions increased numerically with language proficiency (L1 > high L2 proficiency > intermediate L2 proficiency), only L1 speakers seemed to be sensitive to the morphological structure of suffixed derived words. In fact, L1 speakers showed no differences, either between transparent and identical prime–target pairs or between the transparent and pseudo-morphological ones (note that the identical and transparent conditions differed from the control conditions: orthographic and semantic). L2 speakers, however, seem to rely more on the orthographic overlap between prime and target while processing complex suffixed derived words. Still, we believe that authors should be cautious when interpreting the influence of L2 frequency in MMP since the number of studies that entered the analysis (especially those with participants with low L2 proficiency) are limited. Additionally, the influence of participants’ L1 knowledge on their respective L2s should be examined in future research. Further masked priming lexical decision studies combined with other techniques, such as event-related potentials and fMRI, may perhaps help to enable us to disentangle the competing explanations of the results observed. 

## Figures and Tables

**Figure 1 brainsci-13-00127-f001:**
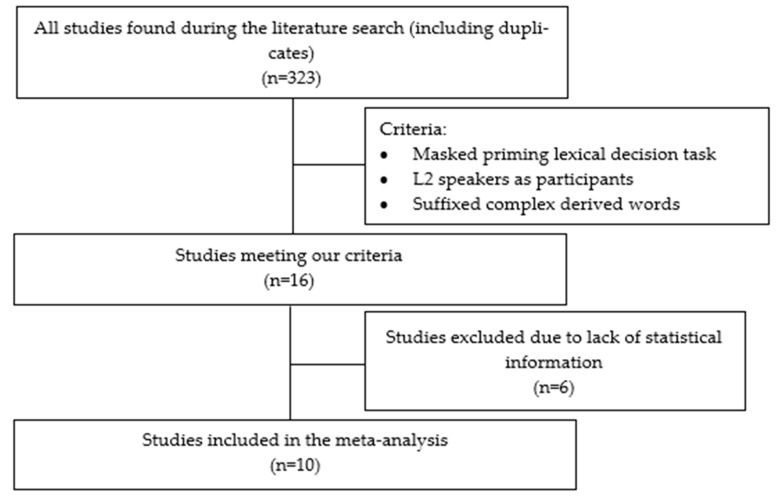
Flow chart of studies included in the meta-analysis.

**Table 1 brainsci-13-00127-t001:** Mean g (SE) [95%CI] for the Prime-Type Relationship for the Full Sample, for Studies with L1 Speakers and with L2 Speakers.

	Identity	Transparent	Pseudo-Derived	Orthographic	Semantic
Full Sample	0.395 (0.023) [0.342, 0.447] *^,a^	0.201 (0.029) [0.139, 0.263] *^,b^	0.102 (0.017) [0.061, 0.143] *^,b,c^	0.075 (0.021) [0.029, 0.12] *^,c,d^	0.015 (0.015) [−0.024, 0.053] ^d^
L1 Speakers	0.407 (0.035) [0.309, 0.506] *^,a^	0.279 (0.023) [0.225, 0.333] *^,a,b^	0.104 (0.028) [0.002, 0.205] *^,b,c^	0.032 (0.019) [−0.014, 0.078] ^c^	0.022 (0.016) [−0.064, 0.108] ^c^
L2 Speakers	0.382 (0.033) [0.288, 0.475] *^,a^	0.118 (0.050) [0.008, 0.229] *^,b^	0.095 (0.014) [0.058, 0.132] *^,b^	0.100 (0.025) [0.042, 0.158] *^,b^	0.010 (0.024) [−0.085, 0.105] ^b^

Note: * indicates that the effect is different from zero. Different superscript letters per row indicate significant differences between conditions. For instance, the effect in a cell with the superscript “a” is significantly different from another effect in the same row with the superscript “b”, and it is not significantly different from another cell in the same row with the superscript “a”. The same rule applies for superscript “c” and “d”, meaning that cells with superscript “c” and “d” significantly differ from each other, and also from cells with superscript “a” and “b”.

## Data Availability

No new data were created or analyzed in this study. Data sharing is not applicable to this article.

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
