# Peer review of "Is There an Early Morphological Decomposition during L2 Lexical Access? A Meta-Analysis on the Morphological Priming Effect"

_brainsci, 2023, doi:10.3390/brainsci13010127_

Round 1

Reviewer 1 Report

This is a very well written article, which presents a rich meta-analysis and an insightful discussion about the role of (derivative) morphology priming effects in native and non-native speakers. I have to say that both the state of the art part and the methodological section are very good and provide very useful information as background for the meta-analysis. I can sincerely say that it is nice when a reviewer has the chance to learn something from a paper under revision.

The article can be published in its present form. There are two minor comments that can be taken into consideration by the authors in order to anticipate some potential doubts by the reader.

1) It is not clear (and I feel that some discussion or clarification is needed) why the authors have chosen those specific 3 keywords (lines 2-3 of section 2). In particular 'bilinguals' does not seem an optimal choice since, even as the authors point out in the discussion, the term could be ambiguous (between balanced and non balanced bilinguals). Something like 'language acquisition' or 'language learning' or simply 'second language' could have provided more results.

2) The role of the morphological type (and its richness of derivation processes) of the native language is not central in most of the considered studies. Intuitively a native speaker of (e.g.) Chinese has a morphological competence different from that of a speaker of (e.g.) Italian or Polish. However, this is a potential problem of the field, and a paper like this is a good chance to state this and discuss its consequences for the final results.

Reviewer 2 Report

This paper is relatively well written. 

However, it needs a proper literature review, which is scattered here and there, all over the paper, between the Introduction section and the section with the methodology. 

That is essential, to 'tidy up' the article and to provide the audience, even the non-specialized one, with a clear set of references and aids. 

A specific section, entitled Literature Review, should be added, with analysis and comments on the works cited and used. 

The sampling can be questionable, at the quantitative level; otherwise, the criteria of selection are, at least, clearly explained. 

The Results section is ok. 

The Discussion should be expanded a little, with more analysis and comments, also in reference to the (used and cited) scientific literature. 

The Introduction is ok, but the Authors should add something about the significance of their paper in the specific field of studies, about their research goals, and on how they plan to achieve them. 

The paper has not a Conclusion, and that is very weird. A Conclusion is necessary, to summarize the findings, and, like in a 'mirror' with the Introduction, to highlight again the research goals of the Authors and to summarize their achievements - this will underline why their paper is significant, in the current panorama of studies. 

The English language is not bad; nonetheless, it can benefit from a re-reading by a native speaker. 

All in all, a quite interesting paper that, in any case, requires a thorough revision, especially at the level of format (and, partly, of contents). 

Thank you very much. 

Round 2

Reviewer 2 Report

The paper has not been revised thoroughly. 

  However, the changes and additions are enough to make it considerable for publication.